# Permutation Diffusion Maps (PDM) with Application to the Image Association Problem in Computer Vision

**Deepti Pachauri**[†]**, Risi Kondor**[§]**, Gautam Sargur**[†]**, Vikas Singh**[‡†]
[†]Dept. of Computer Sciences, University of Wisconsin–Madison
[‡]Dept. of Biostatistics & Medical Informatics, University of Wisconsin–Madison
[§]Dept. of Computer Science and Dept. of Statistics, The University of Chicago
pachauri@cs.wisc.edu  risi@uchicago.edu  gautam@cs.wisc.edu
vsingh@biostat.wisc.edu

## Abstract

Consistently matching keypoints across images, and the related problem of finding clusters of nearby images, are critical components of various tasks in Computer Vision, including Structure from Motion (SfM). Unfortunately, occlusion and large repetitive structures tend to mislead most currently used matching algorithms, leading to characteristic pathologies in the final output. In this paper we introduce a new method, Permutations Diffusion Maps (PDM), to solve the matching problem, as well as a related new affinity measure, derived using ideas from harmonic analysis on the symmetric group. We show that just by using it as a preprocessing step to existing SfM pipelines, PDM can greatly improve reconstruction quality on difficult datasets.

## 1  Introduction

Structure from motion (SfM) is the task of jointly reconstructing 3D scenes and camera poses from a set of images. Keypoints or features extracted from each image provide correspondences between pairs of images, making it possible to estimate the relative camera pose. This gives rise to an association graph in which two images are connected by an edge if they share a sufficient number of corresponding keypoints, and the edge itself is labeled by the estimated matching between the two sets of keypoints. Starting with these putative image to image associations, one typically uses the so-called bundle adjustment procedure to simultaneously solve for the global camera pose parameters and 3-D scene locations, incrementally minimizing the sum of squares of the re-projection error.

Despite their popularity, large scale bundle adjustment methods have well known limitations. In particular, given the highly nonlinear nature of the objective function, they can get stuck in bad local minima. Therefore, starting with a good initial matching (i.e., an informative image association graph) is critical. Several papers have studied this behavior in detail [1], and conclude that if one starts the numerical optimization from an incorrect "seed" (i.e., a subgraph of the image associations), the downstream optimization is unlikely to ever recover.

Similar challenges arise commonly in other fields, ranging from machine learning [2] to computational biology. For instance, consider the *de novo* genome assembly problem in computational biology [3]. The goal here is to reconstruct the original DNA sequence from fragments without a reference genome. Because the genome may have many repeated structures, the alignment problem becomes very hard. In general, reconstruction algorithms start with two maximally overlapping sequences and proceed by selecting the next fragment using a similar criterion. This procedure runs into the same type of issues as described above [4]. It will be useful to have a model that reasons globally over all pairwise information to provide a more robust metric for association. The efficacy of global reasoning will largely depend on the richness of the representation used for encoding pu-

tative pairwise information. The choice of representation is specific to the underlying application, so in this paper, to make our presentation as concrete as possible, we restrict ourselves to describing and evaluating our global association algorithm in the context of the structure from motion problem.

In large scale structure from motion, several authors [5, 6, 7] have recently identified situations where setting up a good image association graph is particularly difficult, and therefore a direct application of bundle adjustment yields highly unsatisfactory results. For example, consider a scene with a large number of duplicate structures (Fig. 1). The preprocessing step in a standard pipeline will match visual features and set up the associations accordingly. A key underlying assumption in most (if not all) approaches is that we observe only a single instance of any structure. This assumption is problematic where scenes have numerous architectural components or recurring patterns, such as windows, bricks, and so on.

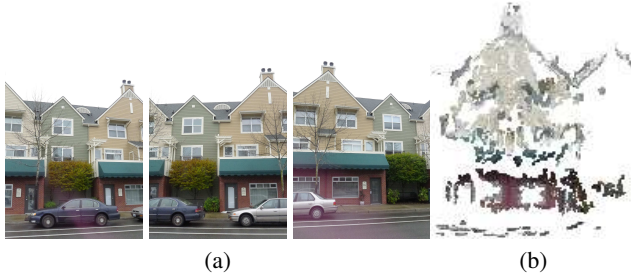

(a)                               (b)

Figure 1: HOUSE sequence. (a) Representative images. (b) Folded reconstruction by traditional SfM pipeline [8, 9].

In Figure 1(a) views that look exactly the same do not necessarily represent the same physical structure. Some (or all) points in one image are actually occluded in the other image. Typical SfM methods will not work well when initialized with such image associations, regardless of which type of solver we use. In our example, the resulting reconstruction will be folded (Figure 1(b)). In other cases [5], we get errors ranging from phantom walls to severely superimposed structures yielding nonsensical reconstructions.

**Related Work.** The issue described above is variously known in the literature as the SfM disambiguation problem or the data/image association problem in structure from motion. Some of the strategies that have been proposed to mitigate it impose additional conditions, such as in [10, 11, 12, 13, 14, 15], but this also breaks down in the presence of large coherent sets of incorrectly matched pairs. One creative solution in recent work is to use metadata alongside images. "Geotags" or GIS data when available have been shown to be very effective in deriving a better initialization for bundle adjustment or as a post-processing step to stitch together different components of a reconstruction. In [6], the authors suggest using image timestamps to impose a natural association among images, which is valuable when the images are acquired by a single camera in a temporal sequence but difficult to deploy otherwise. Separate from the metadata approach, in controlled scenes with relatively less occlusion, missing correspondences yield important local cues to infer potentially incorrect image pairs [6, 7]. Very recently, [5] formalized the intuition that incorrect feature correspondences result in anomalous structures in the so-called visibility graph of the features. By looking at a measure of local track quality (from local clustering), one can reason about which associations are likely to be erroneous. This works well when the number of points is very large, but the authors of [5] acknowledge that for datasets like those shown in Fig. 1, it may not help much.

In contrast to the above approaches, a number of recent algorithms for the association (or disambiguation) problem argue for *global* geometric reasoning. In [16], the authors used the number of point correspondences as a measure of certainty, which was then globally optimized to find a maximum-weight set of consistent pairwise associations. The authors in [17] seek consistency of epipopolar geometry constraints for triplets, whereas [18] expands it over larger consistent cliques. The procedure in [16] takes into account loops of associations concurrently with a minimal spanning tree over image to image matches. In summary, the bulk of prior work suggests that locally based statistics over chained transformations will run into problems if the inconsistencies are more global in nature. However, even if the objectives used are global, *approximate* inference is not known to be robust to *coherent* noise which is exactly what we face in the presence of duplicate structures [19].

**This paper.** If we take the idea of reasoning globally about association consistency using triples or higher order loops to an extreme, it implies deriving the likelihood of a specific image to image association conditioned on *all* other associations. The maximum likelihood expression does not fac-

tor out easily and explicit enumeration quickly becomes intractable. Our approach will make the *group* structure of image to image relationships explicit. We will also operate on the association graph derived from image pairs but with a key distinguishing feature. The association relationships will now be denoted in terms of a 'certificate', that is, the *transformation* which justifies the relationship. The transformation may denote the pose parameters derived from the correspondences or the matching (between features) itself. Other options are possible — as long as this transformation is a *group action* from one set to the other. If so, we can carry over the intuition of consistency over larger cliques of images desired in existing works and rewrite those ideas as invariance properties of functions defined on the *group*. As an example, when the transformation is a matching, each edge in the graph is a permutation, i.e., a member of the symmetric group, $\mathbb{S}_n$. It follows then that a special form of the Laplacian of this graph, derived from the representation theory of the group under consideration, encodes the symmetries of the functions on the group.

The **key contribution** of this paper is to show that the global inference desired in many existing works falls out nicely as a diffusion process using such a Laplacian. We show promising results demonstrating that for various difficult datasets with large repetitive patterns, results from a simple decomposition procedure are, in fact, competitive with those obtained using sophisticated optimization schemes with/without metadata. Finally, we note that the proposed algorithm can either be used standalone to derive meaningful inputs to a bundle adjustment procedure or as a pre-conditioner to other approaches (especially, ones that incorporate timestamps and/or GPS data).

## 2  Synchronization

Consider a collection of $m$ images $\{\mathcal{I}_1, \mathcal{I}_2, \ldots, \mathcal{I}_m\}$ of the same object or scene taken from different viewpoints and possibly under different conditions, and assume that a keypoint detector has detected exactly $n$ landmarks (keypoints) $\{x_1^i, x_2^i, \ldots, x_n^i\}$ in each $\mathcal{I}_i$. Given two images $\mathcal{I}_i$ and $\mathcal{I}_j$, the landmark matching problem consists of finding pairs of landmarks $x_p^i \sim x_p^j$ in the two images which correspond to the same physical feature. This is a critical component of several classical computer vision tasks, including structure from motion.

Assuming that both images contain exactly the same $n$ landmarks, the matching between $\mathcal{I}_i$ and $\mathcal{I}_j$ can be described by a permutation $\tau_{ji} \colon \{1, 2, \ldots, n\} \to \{1, 2, \ldots, n\}$ under which $x_p^i \sim x_{\tau_{ji}(p)}^j$. An initial guess for the $\tau_{ji}$ matchings is usually provided by local image features, such as SIFT descriptors. However, these matchings individually are very much prone to error, especially in the presence of occlusion and repetitive structures. A major clue to correcting these errors is the constraint that matchings must be consistent, i.e., if $\tau_{ji}$ tells us that $x_p^i$ corresponds to $x_q^j$, and $\tau_{kj}$ tells us that $x_q^j$ corresponds to $x_r^k$, then the permutation $\tau_{ki}$ between $\mathcal{I}_i$ and $\mathcal{I}_k$ must assign $x_p^i$ to $x_r^k$. Mathematically, this is a reflection of the fact that if we define the product of two permutations $\sigma_1$ and $\sigma_2$ in the usual way as

$$\sigma_3 = \sigma_2 \sigma_1 \qquad \Longleftrightarrow \qquad \sigma_3(i) = \sigma_2(\sigma_1(i)) \qquad i = 1, 2, \ldots, n,$$

then the $n!$ different permutations of $\{1, 2, \ldots, n\}$ form a group. This group is called the symmetric group of order $n$ and denoted $\mathbb{S}_n$. In group theoretic notation, the consistency conditions require that for any $\mathcal{I}_i, \mathcal{I}_j, \mathcal{I}_k$, the relative matchings between them satisfy $\tau_{kj} \tau_{ji} = \tau_{ki}$. An equivalent condition is that to each $\mathcal{I}_i$ we can associate a base permutation $\sigma_i$ so that $\tau_{ji} = \sigma_j \sigma_i^{-1}$ for any $(i, j)$ pair. Thus, the problem of finding a consistent set of $\tau_{ji}$'s reduces to that of finding just $m$ base permutations $\sigma_1, \ldots, \sigma_m$.

Problems of this general form, where given some (finite or continuous) group $G$, one must estimate a matrix $(g_{ji})_{j,i=1}^m$ of group elements obeying consistency relations, are called synchronization problems. Starting with the seminal work of Singer et al. [20] on synchronization over the rotation group for aligning images in cryo-EM, followed by synchronization over the Euclidean group [21], and most recently synchronization over $\mathbb{S}_n$ for matching landmarks [22][23], problems of this form have recently generated considerable interest.

### 2.1  Vector Diffusion Maps

In the context of synchronizing three dimensional rotations for cryo-EM, Singer and Wu [24] have proposed a particularly elegant formalism, called Vector Diffusion Maps, which conceives of syn-

chronization as diffusing the base rotation $Q_i$ from each image to its neighbors. However, unlike in ordinary diffusion, as $Q_i$ diffuses to $\mathcal{I}_j$, the observed $O_{ji}$ relative rotation of $\mathcal{I}_j$ to $\mathcal{I}_i$ changes $Q_i$ to $O_{ji}Q_i$. If all the $(O_{ji})_{i,j}$ observations were perfectly synchronized, then no matter what path $i \to i_1 \to i_2 \to \ldots \to j$ we took from $i$ to $j$, the resulting rotation $O_{j,i_p}\ldots O_{i_2,i_1}O_{i_1,i}Q_i$ would be the same. However, if some (in many practical cases, the majority) of the $O_{ji}$'s are incorrect, then different paths from one vertex to another contribute different rotations that need to be averaged out. A natural choice for the loss that describes the extent to which the $Q_1,\ldots,Q_m$ imputed base rotations (playing the role of the $\sigma_i$'s in the permutation case) satisfy the $O_{ji}$ observations is

$$\mathcal{E}(Q_1,\ldots,Q_m) = \sum_{i,j=1}^m w_{ij}\| Q_j - O_{ji}Q_i \|_{\text{Frob}}^2 = \sum_{i,j=1}^m w_{ij}\| Q_jQ_i^\top - O_{ji} \|_{\text{Frob}}^2, \qquad (1)$$

where the $w_{ij}$ edge weight descibes our confidence in the rotation $O_{ji}$. A crucial observation is that this loss can be rewritten in the form $\mathcal{E}(Q_1,\ldots,Q_m) = V^\top \mathcal{L}V$, where

$$V = \begin{pmatrix} Q_1 \\ \vdots \\ Q_m \end{pmatrix}, \qquad \mathcal{L} = \begin{pmatrix} d_i\,I & -w_{21}\,O_{21} & \ldots & -w_{m1}\,O_{m1} \\ \vdots & \ddots & & \vdots \\ -w_{1m}\,O_{1m} & -w_{2m}\,O_{2m} & \ldots & d_m\,I \end{pmatrix}, \qquad (2)$$

and $d_i = \sum_{j\neq i} w_{ij}$. Note that since $w_{ij} = w_{ji}$, and $O_{ij} = O_{ji}^{-1} = O_{ji}^\top$, the matrix $\mathcal{L}$ is symmetric. Furthermore, the above is exactly analogous to the way in which in spectral graph theory, (see, e.g.,[25]) the functional $\mathcal{E}(f) = \sum_{i,j} w_{i,j}(f(i) - f(j))^2$ describing the "smoothness" of a function $f$ defined on the vertices of a graph with respect to the graph topology can be written as $f^\top Lf$ in terms of the usual graph Laplacian

$$L_{i,j} = \begin{cases} -w_{i,j} & i \neq j \\ \sum_{k\neq i} w_{i,k} & i = j \end{cases}.$$

The consequence of the latter is that (constraining $f$ to have unit norm and excluding constant functions), the function minimizing $\mathcal{E}(f)$ is the eigenvector of $L$ with (second) smallest eigenvalue. Analogously, in synchronizing rotations, the steady state of the diffusion system, where (1) is minimal, can be computed by forming $V$ from the 3 lowest eigenvalue eigenvectors of $\mathcal{L}$, and then identifying $Q_i$ with $V(i)$, by which we denote its $i$'th $3 \times 3$ block. The resulting consistent array $(Q_jQ_i^\top)_{i,j}$ of imputed relative rotations minimizes the loss (1).

## 3 Permutation Diffusion

Its elegance notwithstanding, the vector diffusion formalism of the previous section seems ill suited for our present purposes of improving the SfM pipeline for two reasons: (1) synchronizing over $\mathbb{S}_n$, which is a finite group, seems much harder than synchronizing over the continuous group of rotations; (2) rather than an actual synchronized array of matchings, what is critical to SfM is to estimate the association graph that captures the extent to which any two images are related to one-another. The main contribution of the present paper is to show that both of these problems have natural solutions in the formalism of group representations.

Our first key observation (already briefly mentioned in [26]) is that the critical step of rewriting the loss (1) in terms of the Laplacian (2) does not depend on any special properties of the rotation group other than the fact (a) rotation matrices are unitary (in fact, orthogonal) (b) if we follow one rotation by another, their matrices simply multiply. In general, for any group $G$, a complex valued function $\rho\colon G \to \mathbb{C}^{d_\rho \times d_\rho}$ which satisfies $\rho(g_2g_1) = \rho(g_2)\rho(g_1)$ is called a representation of $G$. The representation is unitary if $\rho(g^{-1}) = (\rho(g))^{-1} = \rho^\dagger$, where $M^\dagger$ denotes the Hermitian conjugate (conjugate transpose) of $M$. Thus, we have the following proposition.

**Proposition 1.** *Let $G$ be any compact group with identity $e$ and $\rho\colon G \to \mathbb{C}^{d_\rho \times d_\rho}$ be a unitary representation of $G$. Then given an array of possibly noisy and unsynchronized group elements, $(g_{ji})_{i,j}$ and corresponding positive confidence weights $(w_{ji})_{i,j}$, the synchronization loss (assuming $g_{ii} = e$ for all $i$)*

$$\mathcal{E}(h_1,\ldots,h_m) = \sum_{i,j=1}^m w_{ji}\,\|\,\rho(h_jh_i^{-1}) - \rho(g_{ji})\,\|_{Frob}^2 \qquad h_1,\ldots,h_m \in G$$

*can be written in the form $\mathcal{E}(h_1, \ldots, h_m) = V^{\dagger}\mathcal{L}V$, where*

$$V = \begin{pmatrix} \rho(h_1) \\ \vdots \\ \rho(h_m) \end{pmatrix}, \qquad \mathcal{L} = \begin{pmatrix} d_i\,I & -w_{21}\,\rho(g_{21}) & \ldots & -w_{m1}\,\rho(g_{m1}) \\ \vdots & \ddots & & \vdots \\ -w_{1m}\,\rho(g_{1m}) & -w_{2m}\,\rho(g_{2m}) & \ldots & d_m\,I \end{pmatrix}. \qquad (3)$$

To synchronize putative matchings between images, we instantiate this proposition with the appropriate unitary representation of the symmetric group. The obvious choice is the so-called defining representation, whose elements are the familiar permutation matrices

$$\rho_{\mathrm{def}}(\sigma) = P(\sigma) \qquad [P(\sigma)]_{p,q} = \begin{cases} 1 & \sigma(q) = p \\ 0 & \text{otherwise,} \end{cases}$$

since the corresponding loss function is

$$\mathcal{E}(\sigma_1, \ldots, \sigma_m) = \sum_{i,j=1}^{m} w_{ji} \| P(\sigma_j\sigma_i^{-1}) - P(\tau_{ji}) \|_{\mathrm{Frob}}^2. \qquad (4)$$

The squared Frobenius norm in this expression simply counts the number of mismatches between the observed but noisy permutations $\tau_{ji}$ and the inferred permutations $\sigma_j\sigma_i^{-1}$. Furthermore, by the results of the previous section, letting $P_i \equiv P(\sigma(i))$ and $\widehat{P}_{ji} \equiv P(\tau_{ji})$ for notational simplicity, (4) can be written in the form $V^{\top}\mathcal{L}V$ with

$$V = \begin{pmatrix} P_1 \\ \vdots \\ P_m \end{pmatrix}, \qquad \mathcal{L} = \begin{pmatrix} d_i\,I & -w_{21}\,\widehat{P}_{21} & \ldots & -w_{m1}\,\widehat{P}_{m1} \\ \vdots & \ddots & & \vdots \\ -w_{1m}\,\widehat{P}_{1m} & -w_{2m}\,\widehat{P}_{2m} & \ldots & d_m\,I \end{pmatrix}, \qquad (5)$$

Therefore, similarly to the rotation case, synchronization over $\mathbb{S}_n$ can be solved by forming $V$ from the first $d_{\rho_{\mathrm{def}}} = n$ lowest eigenvectors of $\mathcal{L}$, and extracting each $P_{\sigma_i}$ from its $i$'th $n \times n$ block. Here we must take a little care because unless the $\tau_{ji}$'s are already synchronized, it is not a priori guaranteed that the resulting block will be a valid permutation matrix. Therefore, analogously to the procedure described in [22], each block $V(i)$ must be first be multiplied by $V(1)^{\top}$, and then a linear assignment procedure used to find the estimated permutation matrix $\widehat{\sigma}_i$. The resulting algorithm we call Synchronization by Permutation Diffusion.

## 4   Uncertain matches and diffusion distance

The obvious limitation of our framework, as described so far, is that it assumes that each keypoint in each image has a single counterpart in every other image. This assumption is far from being satisfied in realistic scenarios due to occlusion, repetitive structures, and noisy detections. Most algorithms, including [23] and [22], deal with this problem simply by setting the $P_{ij}$ entry of the Laplacian matrix in (5) equal to a weighted sum of all possible permutations. For example, if landmarks number $1 \ldots 20$ are present in both images, but landmarks $21 \ldots 40$ are not, then the effective $P_{ij}$ matrix will have a corresponding $20 \times 20$ block of all ones in it, rescaled by a factor of $1/20$. The consequence of this approach is that each block of the $V$ matrix derived from $\mathcal{L}$ by eigendecomposition will also correspond to a distribution over base permutations.

In principle, this amounts to replacing the single observed matching $\tau_{ji}$ by an appropriate *distribution* $t_{ji}(\tau)$ over possible matchings, and concomitantly replacing each $\sigma_i$ with a distribution $p_i(\sigma)$. However, if some set of landmarks $\{u_1, \ldots, u_k\}$ are occluded in $\mathcal{I}_i$, then each $t_{ji}$ will be agnostic with respect to the assignment of these landmarks, and therefore $p_i$ will be invariant to what labels are assigned to them. Defining $\mu_{u_1 \ldots u_k}$ as any permutation that maps $1 \mapsto u_1, \ldots, k \mapsto u_k$, and regarding $\mathbb{S}_k$ as the subgroup of permutations that permute $1, 2, \ldots, k$ amongst themselves but leave $k+1, \ldots, n$ fixed, any set of permutations of the form $\{\mu_{u_1 \ldots u_k}\gamma\,\nu \mid \gamma \in \mathbb{S}_k\}$ for some $\nu \in \mathbb{S}_n$ is called a right $\mathbb{S}_k$–coset, and is denoted $\mu_{u_1 \ldots u_k}\mathbb{S}_k\nu$. If $\{u_1, \ldots, u_k\}$ are occluded in $\mathcal{I}_i$, then $p_i$ is constant on each $\mu_{u_1 \ldots u_k}\mathbb{S}_k\nu$ (i.e., for any choice of $\nu$).

Whenever there is occlusion, such invariances will spontaneously appear in the $V$ matrix formed from the eigenvectors, and since they are related to which set of landmarks are hidden or uncertain, the invariances are an important clue about the viewpoint that the image was taken from. An affinity

score based on this information is sometimes even more valuable than the synchronized matchings themselves.

The invariance structure of $p_i$ can be read off easily from its so-called autocorrelation function

$$a_i(\sigma) = \sum_{\mu \in \mathbb{S}_n} p_i(\sigma\mu)\, p_i(\mu). \tag{6}$$

In particular, if $\sigma$ is in the coset $\mu_{u_1\ldots u_k}\mathbb{S}_k\mu_{u_1\ldots u_k}^{-1}$, then whatever $\mu$ is, $\sigma\mu$ will fall in the same $\mu_{u_1\ldots u_k}\mathbb{S}_k\nu$ coset, so for any such $\sigma$, $a_i(\sigma) = \sum_{\mu \in \mathbb{S}_n} p_i(\mu)^2$, which is the maximum value that $a_i$ can attain. However, $W(i) := V(i)V(1)^\top$ only reveals a weighted sum $\widehat{p}_i(\rho) := \sum_{\sigma \in \mathbb{S}_n} p_i(\sigma)\rho(\sigma) = W(i)$, rather than the full function $p_i$, so we cannot compute (6) directly.

Recent years have seen the emergence of a number of applications of Fourier transforms on the symmetric group, which, given a function $f\colon \mathbb{S}_n \to \mathbb{R}$, is defined

$$\widehat{f}(\lambda) = \sum_{\sigma \in \mathbb{S}_n} f(\sigma)\,\rho_\lambda(\sigma), \qquad \lambda \vdash n,$$

where the $\rho_\lambda$ are special, so-called irreducible, representations of $\mathbb{S}_n$, indexed by the $\lambda$ integer partitions. Due to space restrictions, we leave the details of this construction to the literature, see, e.g., [27, 28, 29]. Suffice to say that while $V(i)$ is not exactly a Fourier component of $p_i$, it can be expressed as a direct sum of Fourier components

$$V(i) = C^\dagger \Big[\bigoplus_{\lambda \in \Lambda} \widehat{p}_i(\lambda)\Big] C$$

for some unitary matrix $C$ that is effectively just a basis transform. One of the properties of the Fourier transform is that if $h$ is the cross-correlation of two functions $f$ and $g$ (i.e., $h(\sigma) = \sum_{\mu \in \mathbb{S}_n} f(\sigma\mu)\,g(\mu)$), then $\widehat{h}(\lambda) = \widehat{f}(\lambda)\widehat{g}(\lambda)^\dagger$. Consequently, assuming that $V(1)$ has been normalized to ensure that $V(1)^\top V(1) = I$,

$$\widehat{a}_i(\rho) := C^\dagger \Big[\bigoplus_{\lambda \in \Lambda} \widehat{a}_i(\lambda)\Big] C = C^\dagger \Big[\bigoplus_{\lambda \in \Lambda} \widehat{p}_i(\lambda)\widehat{p}_i(\lambda)^\dagger\Big] C = (V(i)V(1))(V(i)V(1))^\top = V(i)V(i)^\top$$

is an easily computable matrix that captures essentially all the coset invariance structure encoded in the inferred distribution $p_i$. To compute an affinity score between some $\mathcal{I}_i$ and $\mathcal{I}_j$ it remains to compare their coset invariance structures, for example, by computing $(\sum_{\sigma \in \mathbb{S}_n} a_i(\sigma)\,a_j(\sigma))^{1/2}$. Omitting certain multiplicative constants arising in the inverse Fourier transform, again using the correlation theorem, one finds that this is equivalent to

$$\Pi(i,j) = \operatorname{tr}\big(V(i)\,V(i)^\top V(j)\,V(j)^\top\big)^{1/2},$$

which we call Permutation Diffusion Affinity (PDA). Remarkably, PDA is closely related to the notion of diffusion similarity derived in [24] for rotations, using entirely different, differential geometric tools. Our experiments show that PDA is surprisingly informative about the actual distance between image viewpoints in physical space, and, as easy it is to compute, can greatly improve the performance of the SfM pipeline.

## 5   Experiments

In our experiments we used Permutation Diffusion Maps to infer the image association matrix of various datasets described in the literature. Geometric ambiguities due to large duplicate structures are evident in each of these datasets, in up to 50% of the matches [6], so even sophisticated SfM pipelines run into difficulties. Our approach is to precede the entire SfM engine with one simple preprocessing step. If our preprocessing step generates good image association information, an existing SfM pipeline which is a very mature software with several linear algebra toolboxes and vision libraries integrated together, can provide good reconstructions. While our primary interest is SfM, to illustrate the utility of PDM, we also present experimental results for scene summarization for a set of images [30]. Additional experiments are available on the project website `http://pages.cs.wisc.edu/~pachauri/pdm/`.

**Structure from Motion (SfM).** We used PDM to generate an image match matrix which is then fed to a state-of-the-art SfM pipeline for 3D reconstruction [8, 9]. As a baseline, we provide these images to a Bundle Adjustment procedure which uses visual features for matching and already has a built-in heuristic outlier removal module. Several other papers have used a similar set of comparisons [6]. For each dataset, SIFT was used to detect and characterize landmarks [31, 32]. We compute putative pairwise matchings $(\tau_{ij})_{i,j=1}^{m}$ by solving $\binom{m}{2}$ linear independent assignments [33] based on their SIFT features. **Image Match Matrix:** Permutation matrix representation is used for putative matchings $(\tau_{ij})_{i,j=1}^{m}$. Here, $n$ is relative large, on the order of 1000. Ideally $n$ is the total number of distinct keypoints in the 3D scene but $n$ is not directly observable. In the experiments, the maximum of keypoints detected across the complete dataset was used to estimate $n$. Eigenvector based procedure computes weighted affinity matrix. While specialized methods can be used to extract a binary image matrix (such that it optimizes a specified criteria), we used a simple thresholding procedure. **3D reconstruction:** We used binary match matrix as an input to a SfM library [8, 9]. Note that we only provide this library the image association hypotheses, leaving all other modules unchanged. With (potentially) good image association information, the SfM modules can sample landmarks more densely and perform bundle adjustment, leaving everything else unchanged. The baseline 3D reconstruction is performed using the same SfM pipeline without intervention.

The **HOUSE** sequence has three instances of similar looking houses, see Figure 1. The diffusion process accumulates evidence and eventually provides strongly connected images in the data association matrix, see Figure 2(a). Warm colors correspond to high affinity between pairs of images. The binary match matrix was obtained by applying a threshold on the weighted matrix, see Figure 2(b). We used this matrix to define the image matching for feature tracks. This means that features are *only* matched between images that are connected in our match matrix. The SfM pipeline was given these image matches as a hypotheses to explain how the images are "connected". The resulting reconstruction correctly gives three houses, see Figure 2(c). The same SfM pipeline when allowed to track features automatically with an outlier removal heuristic, resulted in a folded reconstruction, see Figure 1(b). One may ask if more specialized heuristics will do better, such as time stamps, as suggested in [6]. However, experimental results in [5] and others, strongly suggest that these datasets still remain challenging.

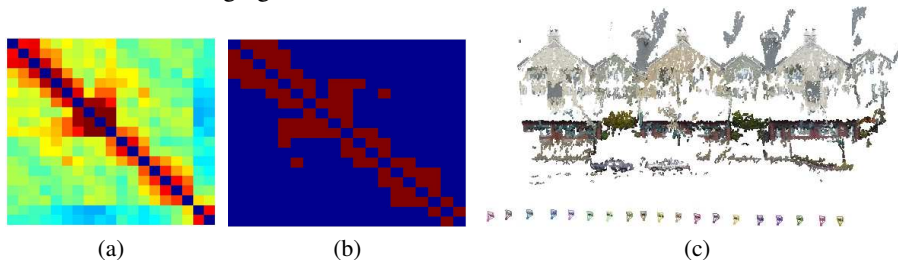

(a)                           (b)                           (c)

Figure 2: House sequence: (a) Weighted image association matrix. (b) Binary image match matrix. (c) PDM dense reconstruction.

The **CUP** dataset has multiple images of a 180 degree symmetric cup from all sides, Figure 3(a). PDM reveals a strongly connected component along the diagonal for this dataset, shown in warm colors in Figure 3(b). Our global reasoning over the space of permutations substantially mitigates coherent errors. The binary match matrix was obtained by thresholding the weighted matrix, see Figure 3(c). As is evident from the reconstructions, the baseline method only reconstruct a "half cup". Due to the structural ambiguity, it also concludes that the cup has two handles, Figure 4(b). PDM reconstruction gives a perfect reconstruction of the "full cup" with one handle as expected, see Figure 4(a). The **OAT** dataset contains two instances of a red oat box, one on the left of the

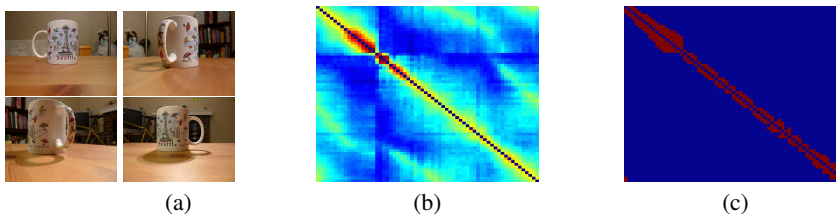

(a)                           (b)                           (c)

Figure 3: (a) Representative images from CUP dataset. (b) Weighted data association matrix. (c) Binary data association matrix.

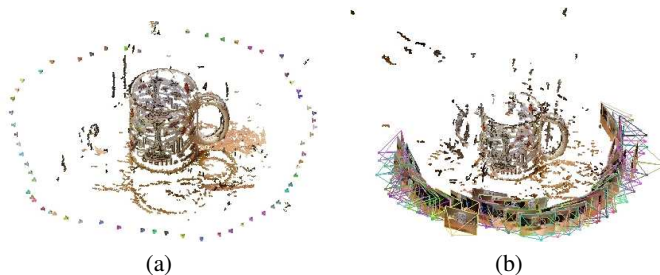

(a)                                                     (b)

Figure 4:  CUP dataset. (a) PDM dense reconstruction. (b) Baseline dense reconstruction.

wheat things, and another on the right, see Figure 5(a). The PDM weighted match matrix and binary match matrix successfully discover strongly connected components, see Figure 5(b-c). The baseline method confused the two oat boxes as one, and reconstructed only a single box, see Figure 6(b). Moreover, the structural ambiguity splits the wheat thins into two pieces. On the other hand, PDM gives a nice reconstruction of the two oat boxes with the entire wheat things in the middle, Figure 6(a). Several more experiments (with videos), can be found on the project website.

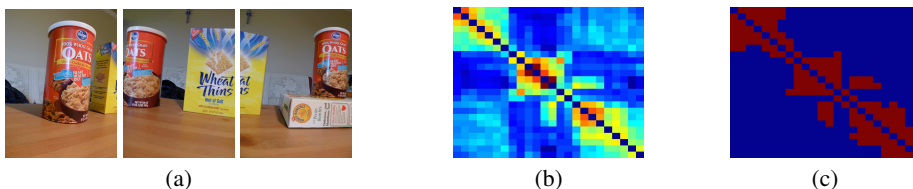

(a)                                   (b)                                   (c)

Figure 5:  (a) Representative images from OAT dataset. (b) Weighted data association matrix. (c) Binary data association matrix.

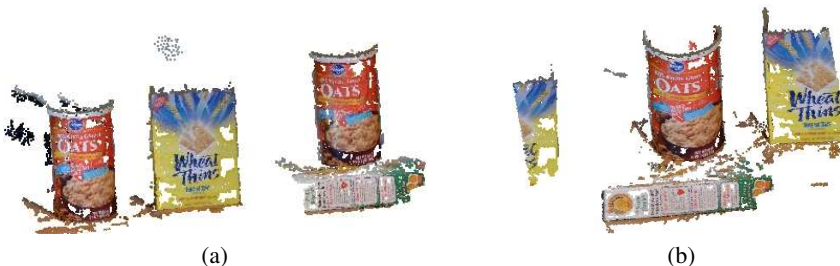

(a)                                                     (b)

Figure 6:  OAT dataset. (a) PDM dense reconstruction. (b) Baseline dense reconstruction.

# 6    Conclusions

Permutation diffusion maps can significantly improve the quality of the correspondences found in image association problems, even when a large number of the initial visual feature matches are erroneous. Our experiments on a variety of challenging datasets from the literature give strong evidence supporting the hypothesis that deploying the proposed formulation, even as a preconditioner, can significantly mitigate problems encountered in performing structure from motion on scenes with repetitive structures. The proposed model can easily generalize to other applications, outside computer vision, involving multi-matching problems.

**Acknowledgments**

This work was supported in part by NSF–1320344, NSF–1320755, and funds from the University of Wisconsin Graduate School. We thank Charles Dyer and Li Zhang for useful discussions and suggestions.

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
