[Reviews · NeurIPS 2014]

Submitted by Assigned_Reviewer_44

This paper applies the diffusion map framework in a permutation setting similar to the cited work [23] which works with rotations in primarily image association problems. The goal is to develop a global association scheme that can better handle the presence of duplicate structures. The authors support their proposed method with experiments on various real image datasets.

This is a clearly written paper and the reviewer expect it to be interesting to the NIPS audience. This reviewer sees the theoretical contribution as the idea of viewing the transformations that need to be coherent as a group and the observation that a preprocessing can bring benefit to existing intricate algorithms and implementations.

However there is a concern also in this regard. One may argue that the idea of thinking in terms of an algebraic group is essentially present already in the cited [23]. The authors describe the case when the transformations are matching and can be viewed as permutations, they derive the formulas but also point out that they are very similar to those of in the rotation setting.

As a result this reviewer is somewhat undecided, but the strong experimental section tipped the balance in favor of recommending acceptance.

A few detailed comments:

The authors should consider citing: Coifman, R.R.; S. Lafon. (2006). "Diffusion maps". Applied and Computational Harmonic Analysis, that introduced diffusion maps.

Line 135: typo in x_p^i \sim x_q^j

Line 150: it may make sense that at the first use of "group" the authors would use "algebraic group" considering the audience and venue (not a group theory conference)

Line 209-210: The Hermitian conjugate is traditionally denoted by M^*. This reviewer sees no reason to deviate from this custom. Also note the transpose is typically marked by capital T, incorrect in (1).

Line 211: Define compact group. Why we need compact?
Summary: This paper has a strong experiment section and presents theory for a compelling idea that warrants publication.

Submitted by Assigned_Reviewer_46

Summary
This paper presents a local feature based multiple image matching algorithm by permutation diffusion maps which is a generalization of vector diffusion maps and derive an image affinity measure from the eigenvectors of diffusion matrix that characteristics the visibility of local features. The derived affinity measure is applied to SfM to resolve the geometric ambiguities of repetitive structures in the image set. The qualitative experiments on three image sets demonstrate the effectiveness of the proposed method.

Pros:
1. The proposed permutation diffusion method for synchronization is novel and theoretically sound.
2. The derived image affinity measure seems to effectively represent the continuous changes of viewpoints, which could help resolve matching ambiguities for repetitive image structures.

Cons:
1. The objective of (4) is to optimize the matchings of local features across multiple images. It will be interesting to evaluate the quality of synchronization directly and to compare with related methods such as [21] in experiments.
2. It is not clear how the affinity measure derived from permutation diffusion matrix compares to other image matching kernels for SfM. For example, one baseline algorithm could be the spatial pyramid matching kernel (Lazebnik et al., CVPR 2006) and another could be spectral graph matching. It will be good if the paper could provide baseline experimental results to better demonstrate the advantages of the proposed affinity measure.
3. More quantitative experiments are needed to evaluate the proposed method for 3D reconstruction.
Summary: The proposed permutation diffusion maps method for image matching is interesting and novel, but more thorough experiments are needed to fully evaluate its strengths and weakness.

Submitted by Assigned_Reviewer_49

The paper proposes an algorithm to solve the matching problem in a global way. The problem arises in the context of Structure from Motion in Computer Vision or in the context of genome assembling problem in computational biology.

The paper introduces the problem focusing on the first scenario and casts it in terms of group theory. Then provides a novel global method for solving the problem by exploiting a group theoretical approach introduced in [19, 23]. Moreover the method is generalized to deal with uncertain matches.

The paper is very clear and the contribution seems original.

It is not clear if the proposed algorithm in Section 3 attains the global minimum of the functional in (4). The approach is quite elegant from a theoretical viewpoint. I suggest to perform more quantitative experiments in order to support the significance of the approach from an empirical perspective.
Summary: The paper deals with the matching problem in computer vision in the context of Structure from Motion. It casts the problem in a group theoretical framework and proposes an elegant algorithm for solving the problem. The paper is well written and the contribution original.
Author Feedback
Author rebuttal: We thank all the reviewers for their careful reading of our work. We address most of the reviewer’s questions below. The revised paper will incorporate all other suggested modifications.

R#1) The authors describe the case when the transformations are matching and can be viewed as permutations, they derive the formulas but also point out that they are very similar to those in the rotation setting [23].

The broad idea of solving synchronization by using group structure and diffusion is already present in [23], but there are crucial differences. The affinity measure in [23] is based on differential geometric ideas such as parallel transport. In contrast, the main contribution of our paper is an affinity derived from the cross-correlation between autocorrelation functions in Fourier form, which is an altogether new idea on any group, together with the analysis in terms of cosets, etc.. This is a fundamental new contribution.

The crucial underlying difference between the two settings is that the rotation group is a continuous Lie group which forms a differentiable manifold, whereas the permutation group is a finite, discrete group, which does not. So ideas from the continuous case cannot just be imported wholesale, even if there are interesting correspondences between some of the formulae.

R#1) Why we need compact?

Compactness lies at the heart of our framework. The irreducible representations of compact groups are always finite dimensional which is key to precisely model the loss function (1).

We thank R#1 for the reference to Coifman et al. We appreciate the minor typo issues highlighted by R#1, we will fix them in the revised version.

R#2) Interesting to evaluate the quality of synchronization directly and to compare with methods such as [21] in experiments.

This is an interesting comment. Our PDM method and the synchronization method of [21] have similar objectives, but there is a subtle yet key distinguishing feature. [21] formulates each pairwise image match as a weighted sum of ALL possible permutations. In the presence of large correlated noise (as opposed to the random noise model of [21]), the pairwise matching solution of [21] is invariably governed by the large coherent erroneous matchings resulting in incorrect pairwise matchings for a fraction of correct match pairs. This is very problematic when the given set of images manifest as clusters with little (or no) shared features across clusters. In fact, we have seen that this behavior is prominent in our early experiments where [21] runs into problems even for “simple” instances. We are happy to include these in the supplement. On the other hand, PDM makes no such assumption. The correlation theorem can well be used to obtain the pairwise matchings which will be robust, by design, to the coherent noise model.

R#2) How the PDM affinity measure compares to other image matching kernels for SfM (the spatial pyramid matching kernel Lazebnik et al., and spectral graph matching). It will be good if the paper could provide baseline experimental results to better demonstrate the advantages of the proposed affinity measure.

We are not sure if we understand the concern here. Therefore, we provide some clarification. Our goal in this paper is NOT to derive a better procedure to obtain image-to-image affinities. In fact, such information (e.g., obtained from Lazebnik et al., or spectral graph matching) serves as the input to our algorithm, not competing alternatives. While we used SIFT features to obtain putative matches in the paper, we could have easily used any other method for this task.

Note that any such method is local in the sense that if there are two images with similar looking structure but do not represent the same physical structure, these methods will conclude a strong affinity for the image pair. Fundamentally, there is no easy “fix” to this problem. As a result, the downstream reconstruction will assume that the image pair corresponds to legitimate neighbors and will produce a folded reconstruction. This paper proposes a method that derives the pairwise affinity measure conditioned on *all other image pairs*. The global reasoning used in this paper is independent of the specific local pairwise image matching procedure one uses. We are happy to provide some example before/after affinity matrices on the paper webpage using different features, as suggested.

R#2, R#3) More quantitative experiments are needed to evaluate the proposed method for 3D reconstruction and to support the significance of the approach from an empirical perspective.

The evaluation procedure for 3D reconstruction is qualitative because for the types of large scale SfM experiments our proposed algorithm seeks to improve, ground truth data is impractical in all (but the most trivial) cases. Our simple proposal is to run an existing deployment of large scale SfM with image-to-image neighborhoods derived from our algorithm. The benefits here are clear: when the upstream method fails, the reconstruction looks completely nonsensical. These are precisely the situations where our method offers radical improvements. We closely modeled our experiments based on various recent papers which dealt with the disambiguation problem in 3D reconstruction [5,6,7]. The datasets we used are considered quite challenging. As noted by R#1, our results are significantly better than [5,6,7] as well as the baseline method.

We can include comparisons of the image match matrix of our method and the baseline method. These plots were left out of the main paper to make space for other compelling qualitative experiments.